# SCALING SUPERVISED LOCAL LEARNING WITH AUGMENTED AUXILIARY NETWORKS

**Chenxiang Ma[1], Jibin Wu[1]\*, Chenyang Si[2], Kay Chen Tan[1]**
[1]The Hong Kong Polytechnic University, Hong Kong SAR, China
[2]Nanyang Technological University, Singapore

## ABSTRACT

Deep neural networks are typically trained using global error signals that back-propagate (BP) end-to-end, which is not only biologically implausible but also suffers from the update locking problem and requires huge memory consumption. Local learning, which updates each layer independently with a gradient-isolated auxiliary network, offers a promising alternative to address the above problems. However, existing local learning methods are confronted with a large accuracy gap with the BP counterpart, particularly for large-scale networks. This is due to the weak coupling between local layers and their subsequent network layers, as there is no gradient communication across layers. To tackle this issue, we put forward an augmented local learning method, dubbed AugLocal. AugLocal constructs each hidden layer's auxiliary network by uniformly selecting a small subset of layers from its subsequent network layers to enhance their synergy. We also propose to linearly reduce the depth of auxiliary networks as the hidden layer goes deeper, ensuring sufficient network capacity while reducing the computational cost of auxiliary networks. Our extensive experiments on four image classification datasets (i.e., CIFAR-10, SVHN, STL-10, and ImageNet) demonstrate that AugLocal can effectively scale up to tens of local layers with a comparable accuracy to BP-trained networks while reducing GPU memory usage by around 40%. The proposed AugLocal method, therefore, opens up a myriad of opportunities for training high-performance deep neural networks on resource-constrained platforms. Code is available at `https://github.com/ChenxiangMA/AugLocal`.

## 1 INTRODUCTION

Artificial neural networks (ANNs) have achieved remarkable performance in pattern recognition tasks by increasing their depth (Krizhevsky et al., 2012; LeCun et al., 2015; He et al., 2016; Huang et al., 2017). However, these deep ANNs are trained end-to-end with the backpropagation algorithm (BP) (Rumelhart et al., 1985), which faces several limitations. One critical criticism of BP is its biological implausibility (Crick, 1989; Lillicrap et al., 2020), as it relies on a global objective optimized by backpropagating error signals across layers. This stands in contrast to biological neural networks that predominantly learn based on local signals (Hebb, 1949; Caporale & Dan, 2008; Bengio et al., 2015). Moreover, layer-by-layer error backpropagation introduces the update locking problem (Jaderberg et al., 2017), where hidden layer parameters cannot be updated until both forward and backward computations are completed, hindering efficient parallelization of the training process.

Local learning (Bengio et al., 2006; Mostafa et al., 2018; Belilovsky et al., 2019; Nøkland & Eidnes, 2019; Illing et al., 2021; Wang et al., 2021) has emerged as a promising alternative for training deep neural networks. Unlike BP, local learning rules train each layer independently using a gradient-isolated auxiliary network with local objectives, avoiding backpropagating error signals from the output layer. Consequently, local learning can alleviate the update locking problem as each layer updates its parameters independently, allowing for efficient parallelization of the training process (Belilovsky et al., 2020; Laskin et al., 2020; Gomez et al., 2022). Additionally, local learning does not require storing intermediate network states, as required by BP, leading to much lower memory consumption (Löwe et al., 2019; Wang et al., 2021; Wu et al., 2021).

---

*Corresponding author: jibin.wu@polyu.edu.hk

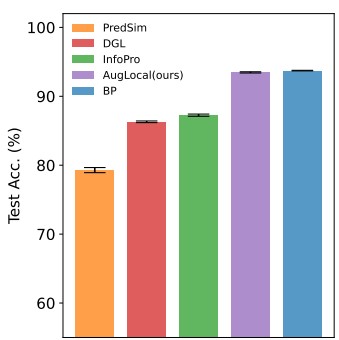

However, local learning suffers from lower accuracy than BP, especially when applied to large-scale networks with numerous independently optimized layers (Belilovsky et al., 2020; Wang et al., 2021; Siddiqui et al., 2023). This is primarily because hidden layers only learn representations that suit their local objectives rather than benefiting the subsequent layers, since there is a lack of feedback interaction between layers. Previous efforts in supervised local learning have focused on designing local losses to provide better local guidance (Nøkland & Eidnes, 2019), such as adding reconstruction loss to preserve input information in InfoPro (Wang et al., 2021). However, these methods still fall short of BP in terms of accuracy, especially for very deep networks (see Figure 1).

Figure 1: Comparison of supervised local learning rules and BP on CIFAR-10 dataset. ResNet-32 architecture, with 16 local layers, has been used in this experiment.

In this paper, we address the scalability issue of supervised local learning methods by strengthening the synergy between local layers and their subsequent layers. To this end, we propose an augmented local learning rule, namely AugLocal, which builds each local layer's auxiliary network using a uniformly sampled small subset of its subsequent layers. To reduce the additional computational cost of auxiliary networks, we further propose a pyramidal structure that linearly decreases the depth of auxiliary networks as the local layer approaches the output. This is motivated by the fact that top layers have fewer subsequent layers than bottom ones. The proposed method has been extensively evaluated on image classification datasets (i.e., CIFAR-10 (Krizhevsky et al., 2009), SVHN (Netzer et al., 2011), STL-10 (Coates et al., 2011), and ImageNet (Deng et al., 2009)), using varying depths of commonly used network architectures. Our key contributions are threefold:

- We propose the first principled rule for constructing auxiliary networks that promote the synergy among local layers and their subsequent layers during supervised local learning. Our method, AugLocal, provides the first scalable solution for large-scale networks with up to 55 independently trained layers.

- We thoroughly validate the effectiveness of AugLocal on a number of benchmarks, where it significantly outperforms existing supervised local learning rules and achieves comparable accuracies to BP while reducing GPU memory usage by around 40%. Our ablation studies further verify that the pyramidal structure can reduce approximately 45% of auxiliary networks' FLOPs while retaining similar levels of accuracy. Furthermore, we demonstrate the generalizability of our approach across various convolutional architectures

- We provide an in-depth analysis of the hidden representations learned by local learning and BP methods. Our results reveal that AugLocal can effectively learn hidden representations similar to BP that approach the linear separability of BP, providing a compelling explanation on the efficacy of the proposed method.

## 2 BACKGROUND: SUPERVISED LOCAL LEARNING

Given a deep neural network $\mathcal{F}_{\boldsymbol{\Theta}} : \mathcal{X} \to \mathcal{Y}$, the forward propagation of an input sample $\boldsymbol{x} \in \mathcal{X}$ can be represented by

$$\mathcal{F}_{\boldsymbol{\Theta}}(\boldsymbol{x}) = g \circ f^L \circ f^{L-1} \circ \cdots \circ f^1(\boldsymbol{x}) \tag{1}$$

where $\circ$ stands for function composition, and $g(\cdot)$ represents the final classifier. $L$ is the number of layers excluding the classifier. $f^\ell(\cdot)$ denotes the operational function at layer $\ell$, $1 \leq \ell \leq L$. Parameters of all trainable layers are denoted by $\boldsymbol{\Theta} = \{\boldsymbol{\theta}^i\}_{i=1}^{L+1}$.

Instead of backpropagating global error signals like BP, supervised local learning employs layer-wise loss functions to update the parameters of each hidden layer (see Figure 2(b)). Specifically, for any hidden layer $\ell$, an auxiliary network $\mathcal{G}_{\boldsymbol{\Phi}}^\ell : \mathcal{H} \to \mathcal{Y}$ maps the hidden representation $\boldsymbol{h}^\ell \in \mathcal{H}$ to the output space. The computation of the layer $\ell$ can be represented by $\boldsymbol{h}^\ell = f^\ell(\text{sg}(\boldsymbol{h}^{\ell-1}))$ where sg stands for the stop-gradient operator that prevents gradients from being backpropagated between adjacent layers. During training, a local loss function $\hat{\mathcal{L}}^\ell(\mathcal{G}_{\boldsymbol{\Phi}}^\ell(\boldsymbol{h}^\ell), \boldsymbol{y})$ is computed based on the

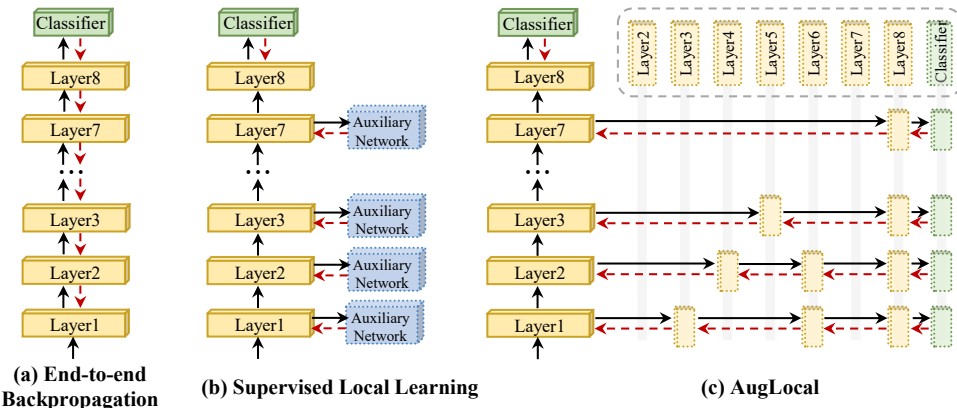

Figure 2: **Comparison of (a) end-to-end backpropagation (BP), (b) supervised local learning, and (c) our proposed AugLocal method.** Unlike BP, supervised local learning trains each hidden layer with a gradient-isolated auxiliary network. AugLocal constructs the auxiliary networks by uniformly selecting a given number of layers from each hidden layer's subsequent layers. Additionally, the depth of auxiliary networks linearly decreases as the hidden layer approaches the final classifier. Black and red arrows represent forward and gradient propagation during training.

auxiliary network output $\mathcal{G}_{\Phi}^{\ell}(\boldsymbol{h}^{\ell})$ and the label $\boldsymbol{y}$, and the parameters of both the layer $\ell$ and its auxiliary network are updated solely based on the gradients derived from the local loss. Note that the last hidden layer $L$ of the primary network (i.e., the network we are interested in) does not require any auxiliary network as it is connected directly to the output classifier. The parameters of layer $L$ and the classifier are updated jointly according to the global loss $\mathcal{L}(\mathcal{F}(\boldsymbol{x}), \boldsymbol{y})$. It is worth mentioning that the auxiliary networks are discarded during inference, and the primary network operates exactly the same as BP-trained ones.

**Supervised local loss** Supervised local learning rules typically adopt the same local loss function as the output layer, such as the cross-entropy (CE) loss used for classification tasks (Belilovsky et al., 2019; 2020). Recent works have proposed incorporating additional terms into the local objective function. For instance, PredSim (Nøkland & Eidnes, 2019) proposes to minimize an additional similarity matching loss to maximize inter-class distance and minimize intra-class distance between pairs. InfoPro (Wang et al., 2021) shows that earlier layers may discard task-relevant information after local learning, which adversely affects the network's final performance. Therefore, they introduce a reconstruction loss to each hidden layer, coupled with the CE loss, to enforce hidden layers to preserve useful input information.

**Auxiliary networks** Existing supervised local learning works use handcrafted auxiliary networks. For example, DGL (Belilovsky et al., 2020) utilizes MLP-SR-aux, which incorporates spatial pooling and three $1\times1$ convolutional layers, followed by average pooling (AP) and a 3-layer fully-connected (FC) network. PredSim (Nøkland & Eidnes, 2019) uses two separate auxiliary networks - a FC layer with AP for the CE loss and a convolutional layer for similarity matching loss. Similarly, InfoPro (Wang et al., 2021) adopts two auxiliary networks comprising a single convolutional layer followed by two FC layers for the CE loss and two convolutional layers with up-sampling operations for reconstruction loss. Despite their promising results under relaxed local learning settings, there is a lack of principled guidelines for designing auxiliary networks that can promote synergy between the local layers and their subsequent layers, as well as can generalize across different networks.

## 3 SUPERVISED LOCAL LEARNING WITH AUGMENTED AUXILIARY NETWORKS

### 3.1 PROBLEM FORMULATION

In supervised local learning, each layer is updated independently with its own local loss function, which can alleviate the update locking problem (Jaderberg et al., 2017; Belilovsky et al., 2020). However, this problem will become exacerbated when the depth of auxiliary networks grows. Therefore,

we impose a maximum depth $d$ (i.e., the number of trainable layers) on auxiliary networks to control the degree of update locking. Note that a smaller value of $d$ can enhance training efficiency, as the theoretical training time ratio between AugLocal and BP is approximately $\frac{d+1}{L+1}$ (refer to Appendix A.1 for detailed analysis). Additionally, we impose a constraint on the computational cost of auxiliary networks to ensure efficient training. With the target of jointly minimizing the output and local losses, the supervised local learning can be formulated as:

$$\min_{\Theta,\mathcal{M}} \quad \mathcal{L}(\mathcal{F}(\boldsymbol{x}),\boldsymbol{y}) + \sum_{\ell=1}^{L-1} \hat{\mathcal{L}}^{\ell}(\mathcal{G}_{\boldsymbol{\Phi}}^{\ell}(f^{\ell}(\text{sg}(\boldsymbol{h}^{\ell-1}))),\boldsymbol{y})$$

$$s.t. \quad \begin{cases} |\mathcal{G}_{\boldsymbol{\Phi}}^{\ell}| \leq d, \quad \ell = 1,\ldots,L-1, \\ \sum_{\ell=1}^{L-1} \text{FLOPs}(\mathcal{G}_{\boldsymbol{\Phi}}^{\ell}(\boldsymbol{h}^{\ell})) \leq \gamma. \end{cases} \tag{2}$$

where $\mathcal{M} = \{\boldsymbol{\Phi}^1,\ldots,\boldsymbol{\Phi}^{L-1}\}$ is the set of parameters for all auxiliary networks. $|\mathcal{G}_{\boldsymbol{\Phi}}^{\ell}|$ denotes the number of trainable layers of the $\ell^{th}$ auxiliary net. $d$ and $\gamma$ represent the given maximum depth limit and maximum floating point operations (FLOPs) for auxiliary nets, respectively.

## 3.2 Augmented auxiliary networks

Following supervised local learning, each hidden layer is updated based on its gradient-isolated auxiliary network, without any feedback about the global loss from top layers. As a result, the optimization of bottom layers tends to prioritize short-term gains which, however, may overlook some essential features that can lead to better performance for subsequent layers. This has been seen in early works (Wang et al., 2021) and our feature representation analysis presented in Section 4.3.

To mitigate the problem of short-sightedness and improve the performance of final outputs, it is desirable to learn similar hidden representations as those developed by BP. To this end, we introduce a structure prior to the local auxiliary networks. In particular, we propose constructing the auxiliary network of a hidden layer by uniformly selecting a given small number of layers from its subsequent layers, as illustrated in Figure 2(c). Our intuition is that by leveraging the structures of subsequent layers from the primary network, we can capture the essential transformations of the subsequent layers, which could guide the hidden layer to generate a more meaningful feature representation that resembles the BP-trained one, and ultimately benefit the final output.

Specifically, for a hidden layer $\ell$, we select layers from the primary network with layer indices $\beta_i = \ell + \lfloor \frac{(L-\ell)i}{d^{\ell}-1} \rceil, i = 1,\ldots,d^{\ell}-1$, where $\lfloor \cdot \rceil$ is the nearest integer function. $d^{\ell}$ denotes the given depth of the $\ell^{th}$ auxiliary network. Then, we construct the auxiliary network using these selected layers, which can be denoted as $\mathcal{G}^{\ell} = g \circ f^{\beta_{d^{\ell}-1}} \circ f^{\beta_{d^{\ell}-2}} \circ \cdots \circ f^{\beta_1}(\boldsymbol{h}^{\ell})$. Here, we use the same classifier $g(\cdot)$ as used in the primary network, which typically consists of a global average pooling layer followed by a fully-connected layer. Note that the selection of the auxiliary network takes place prior to training, and only the architectural structures of the selected layers are employed in the construction of the auxiliary network. When the dimensions of two adjacent layers in the auxiliary network do not match, we modify the input channel number of the later layer to match the output channel number of the preceding one. Furthermore, we perform downsampling when the channel number of the later layer is doubled or increased even further.

## 3.3 Pyramidal depth for auxiliary networks

Since each hidden layer is coupled with an auxiliary network, which will introduce substantial computational costs during the training process. To alleviate this challenge, we propose a method for reducing the computational burden associated with auxiliary networks by shrinking the depth of auxiliary networks progressively. We refer to this approach as pyramidal depth for auxiliary networks, which linearly reduces the depth of auxiliary networks as the hidden layer index increases. Our approach is motivated by the fact that top layers in the primary network have fewer subsequent layers than the lower ones. Specifically, we obtain the depth $d^{\ell}$ of the $\ell^{th}$ auxiliary network as per:

$$d^{\ell} = \min\left(\left\lfloor (1-\tau\frac{\ell-1}{L-2})d + \tau\frac{\ell-1}{L-2}d_{\min} \right\rceil, L-\ell+1\right), \quad \ell = 1,\ldots,L-1 \tag{3}$$

This ensures that $d^{\ell}$ starts at a predefined maximum depth $d$ and asymptotically approaches a minimum depth $d_{\min}$ with the layer $\ell$ going deeper. The decay rate is determined by the factor $\tau$.

Table 1: Comparison of supervised local learning methods and BP on image classification datasets. The averaged test accuracies and standard deviations are reported from three independent trials. $L$ denotes the number of independently trained layers (residual blocks in ResNets). $d$ represents the depth of auxiliary networks. For example, $d = 2$ indicates that each auxiliary network contains one hidden layer in addition to the linear classifier.

| Network | Method | CIFAR-10 | SVHN | STL-10 |
|---------|--------|----------|------|--------|
| ResNet-32 $(L = 16)$ | BP | 93.73±0.04 | 97.01±0.03 | 80.80±0.17 |
| | PredSim (Nøkland & Eidnes, 2019) | 79.29±0.37 | 92.02±0.35 | 70.67±0.39 |
| | InfoPro (Wang et al., 2021) | 87.26±0.16 | 93.30±0.73 | 70.85±0.14 |
| | DGL (Belilovsky et al., 2020) | 86.30±0.12 | 95.14±0.09 | 73.13±1.08 |
| | AugLocal ($d = 2$) | 91.12±0.24 | 95.86±0.04 | 78.58±0.66 |
| | AugLocal ($d = 3$) | 92.26±0.20 | 96.43±0.02 | 79.79±0.27 |
| | AugLocal ($d = 4$) | 93.08±0.10 | 96.79±0.08 | 80.65±0.16 |
| | AugLocal ($d = 5$) | 93.38±0.11 | **96.87±0.03** | 80.73±0.18 |
| | AugLocal ($d = 6$) | **93.47±0.09** | 96.85±0.08 | **80.95±0.55** |
| ResNet-110 $(L = 55)$ | BP | 94.61±0.18 | 97.10±0.05 | 80.41±0.74 |
| | PredSim (Nøkland & Eidnes, 2019) | 74.95±0.36 | 89.90±0.76 | 68.91±0.48 |
| | InfoPro (Wang et al., 2021) | 86.95±0.46 | 92.26±0.59 | 70.61±0.50 |
| | DGL (Belilovsky et al., 2020) | 85.69±0.32 | 95.12±0.06 | 72.27±0.51 |
| | AugLocal ($d = 2$) | 90.98±0.05 | 95.92±0.10 | 78.29±0.37 |
| | AugLocal ($d = 3$) | 92.62±0.22 | 96.45±0.08 | 79.30±0.26 |
| | AugLocal ($d = 4$) | 93.22±0.17 | 96.74±0.11 | **80.77±0.33** |
| | AugLocal ($d = 5$) | 93.75±0.20 | 96.85±0.05 | 80.20±0.56 |
| | AugLocal ($d = 6$) | **93.96±0.15** | **96.96±0.01** | 80.40±0.17 |

Table 2: Comparison of different learning methods on VGG19 with 16 independently trained layers.

| | BP | PredSim | DGL | AugLocal ($d = 2$) | AugLocal ($d = 3$) | AugLocal ($d = 4$) |
|---|-----|---------|-----|------|------|------|
| CIFAR-10 | 93.68±0.11 | 86.49±0.30 | 92.48±0.06 | 93.16±0.28 | 93.46±0.19 | **93.69±0.05** |
| SVHN | 96.79±0.08 | 94.07±0.43 | 96.63±0.08 | 96.48±0.07 | **96.65±0.04** | 96.64±0.06 |

## 4 EXPERIMENTS

### 4.1 EXPERIMENTAL SETUP

We evaluate the performance of our method on four datasets: CIFAR-10 (Krizhevsky et al., 2009), SVHN (Netzer et al., 2011), STL-10 (Coates et al., 2011), and ImageNet (Deng et al., 2009). Experiments are primarily based on ResNets (He et al., 2016) and VGGs (Simonyan & Zisserman, 2014) with varying depths. We compare AugLocal with BP and three state-of-the-art supervised local learning methods, including DGL (Belilovsky et al., 2020), PredSim (Nøkland & Eidnes, 2019), and InfoPro (Wang et al., 2021). For a fair comparison, these baseline methods are re-implemented in PyTorch using consistent training settings (see details in Appendix A.2). We follow the original configurations as stated in their respective papers for the auxiliary networks of DGL, PredSim, and InfoPro. The structures and computational costs of these auxiliary networks are detailed in Appendix A.3. In addition, local learning rules are evaluated with networks being divided into independently trained local layers, which is the minimal indivisible unit. For example, local layers are residual blocks in ResNets and convolutional layers in VGGs. We adopt the simultaneous training scheme for local learning rules, which sequentially triggers the training process of each local layer with every mini-batch of training data (Belilovsky et al., 2020; Nøkland & Eidnes, 2019; Wang et al., 2021). For AugLocal, we begin with the maximum depth limit of $d = 2$, whereby each auxiliary network containing one hidden layer in addition to the linear classifier. Considering the varying availability of computational resources in different scenarios, we progressively increase the depth limit to 6 in our experiments, reporting the results for each value. We adopt the cross-entropy loss as the local loss function of AugLocal and the decay rate $\tau = 0.5$ in experiments.

### 4.2 RESULTS ON IMAGE CLASSIFICATION DATASETS

**Results on various image classification benchmarks** We begin our evaluation by comparing AugLocal against baseline methods on CIFAR-10, SVHN and STL-10 datasets. Here, we use ResNet-

Table 3: Results on the validation set of ImageNet.

| Network | Method | Top-1 Acc. | Top-5 Acc. |
|---------|--------|-----------|-----------|
| VGG13 ($L = 10$) | BP | 71.59 | 90.37 |
| | DGL | 67.32 | 87.81 |
| | AugLocal | **70.92** | **90.13** |
| ResNet-34 ($L = 17$) | BP | 74.28 | 91.76 |
| | AugLocal | **73.95** | **91.70** |
| ResNet-101 ($L = 34$) | BP | 77.34 | 93.71 |
| | AugLocal | **76.70** | **93.29** |

Table 4: Comparison of GPU memory usage, measured by those best-performing models on each network and dataset with a batch size of 1024. 'GC' refers to gradient checkpointing (Chen et al., 2016).

| Dataset | Network | Method | GPU Memory (GB) |
|---------|---------|--------|-----------------|
| CIFAR-10 | ResNet-32 ($L = 16$) | BP | 3.15 |
| | | AugLocal | 1.67 (↓ **47.0%**) |
| | ResNet-110 ($L = 55$) | BP | 9.27 |
| | | GC | 3.03 (↓ 67.3%) |
| | | AugLocal | 1.72 (↓ **81.5%**) |
| ImageNet | ResNet-34 ($L = 17$) | BP | 42.95 |
| | | AugLocal | 29.04 (↓ **32.4%**) |
| | ResNet-101 ($L = 34$) | BP | 157.12 |
| | | AugLocal | 97.65 (↓ **37.9%**) |

32 and ResNet-110 that have 16 and 55 local layers, respectively. As shown in Table 1, all three supervised local learning rules exhibit significantly lower accuracy compared to BP, and this accuracy gap increases as the number of layers increases. In contrast, our AugLocal outperforms all local learning methods with only a single layer and a linear classifier in its auxiliary networks ($d = 2$). This observation indicates our auxiliary network design strategy is more effective than those handcrafted designs as deeper auxiliary networks are used in DGL (Belilovsky et al., 2020) and InfoPro (Wang et al., 2021). As expected, the accuracy of AugLocal improves further with deeper auxiliary networks. This is because the local learning can better approximate the global learning when more layers of the primary network are selected into the construction of auxiliary networks. Notably, AugLocal achieves comparable accuracies to BP using auxiliary networks with no more than six layers on both network architectures.

Moreover, we also conduct experiments with VGG19 architecture, which consists of 16 independently trained layers. As shown in Table 2, AugLocal consistently outperforms local learning baselines and achieves comparable or better accuracy to BP. This reaffirms the effectiveness of AugLocal in handling deeper networks and its ability to approximate the learning achieved by BP.

We further evaluate the generalization ability of AugLocal on three popular convolutional networks, including MobileNet (Sandler et al., 2018), EfficientNet (Tan & Le, 2019), and RegNet (Radosavovic et al., 2020). These architectures differ significantly from VGGs and ResNets, both in terms of their overall structures and the types of building blocks. Our results in Appendix A.4 demonstrate that AugLocal consistently obtains comparable accuracy to BP, regardless of the network structure, highlighting the potential of AugLocal to generalize across different network architectures.

**Results on ImageNet**  We validate the effectiveness of AugLocal on ImageNet (Deng et al., 2009) using three networks with varying depths. Specifically, we adopt VGG13 (Simonyan & Zisserman, 2014), ResNet-34 (He et al., 2016), and ResNet-101 (He et al., 2016), which contain 10, 17, and 34 local layers. It is worth noting that the majority of existing supervised local learning rules have not been extensively evaluated on ImageNet using the challenging layer-wise local learning setting.

As shown in Table 3, our AugLocal achieves comparable top-1 accuracy to BP on VGG13 ($d = 3$), which significantly outperforms DGL (Belilovsky et al., 2020) by 3.6%. In the case of deeper architectures, such as ResNet-34 ($d = 5$) and ResNet-101 ($d = 4$), AugLocal consistently approaches the same level of accuracies as that of BP. These results demonstrate the effectiveness of AugLocal in scaling up to large networks with tens of independently trained layers on the challenging ImageNet.

### 4.3 REPRESENTATION SIMILARITY ANALYSIS

We have shown that AugLocal can achieve comparable accuracy to BP across different networks. However, a critical question remains: how does AugLocal achieve such high performance? Does AugLocal develop identical hidden representations as BP or different ones? We provide an in-depth analysis by studying representation similarity and linear separability between AugLocal and BP.

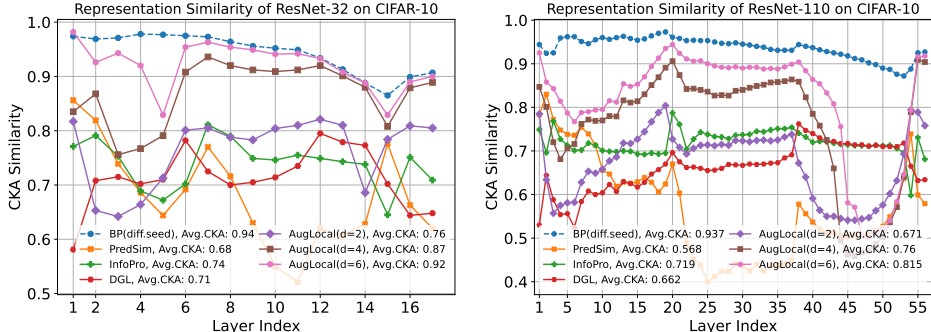

Figure 3: Comparison of layer-wise representation similarity. We utilize centered kernel alignment (CKA) (Kornblith et al., 2019) to measure the layer-wise similarity of representations between BP and other local learning rules. To provide a fair baseline for BP, we measure the similarity between two networks trained with different random seeds. The average CKA similarity scores for different learning rules are provided in the legend.

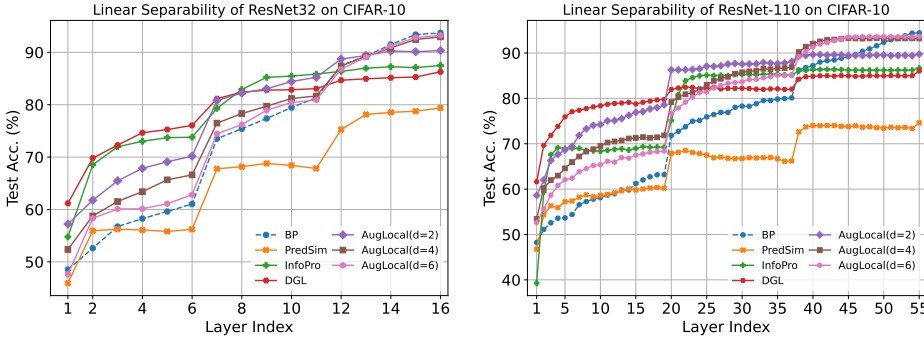

Figure 4: Comparison of layer-wise linear separability across different learning rules.

**Representation similarity** We use centered kernel alignment (CKA) (Kornblith et al., 2019) to quantitatively analyze the representation similarity between BP and local learning rules. Specifically, we calculate the CKA similarity of each local layer, and then we compute an average CKA score based on all local layers.

Figure 3 presents the results of ResNet-32 (He et al., 2016) and ResNet-110 (He et al., 2016). We observe a monotonic increase in similarity scores as the depth of AugLocal auxiliary networks increases. This suggests that hidden layers trained with AugLocal generate representations that progressively resemble those generated by BP as the depth of auxiliary networks increases. This phenomenon provides a compelling explanation for AugLocal's ability to achieve comparable accuracies to BP. We notice that earlier layers achieve lower similarity scores. This is expected because the features of earlier layers are less specific and highly redundant as compared to top layers. However, despite the lower similarity scores, the earlier layers still play a significant role in providing meaningful features that contribute to the overall performance of the network. This is evident from the high representation similarity observed between the middle and final output layers.

**Linear probing** We further employ the linear probing technique to compare the hidden layer linear separability of different learning methods. Specifically, we freeze the pre-trained parameters of the primary network and train additional linear classifiers attached to each hidden layer. As shown in Figure 4, our results reveal that most local learning baselines, when compared to BP, achieve higher accuracies at earlier layers but perform significantly worse at middle and output layers. This indicates that these local learning rules tend to be short-sighted, with early features predominantly optimized for their local objectives rather than supporting the subsequent layers (Wang et al., 2021). In contrast, AugLocal exhibits a poorer linear separability in the early layers, suggesting the learned features are more general. However, a significant improvement in linear separability is observed in middle and output layers, closely following the pattern achieved by BP. This trend is more obvious as the depth of auxiliary nets increases. These results indicate that AugLocal can alleviate the short-sightedness problem of other local learning methods, providing more useful features for top layers.

Table 5: Comparison of different auxiliary networks construction strategies with ResNet-110 on CIFAR-10. Unif., Seq., and Repe. represent the uniform, sequential, and repetitive strategies. C1×1 and C3×3 denote constructing auxiliary networks using 1×1 and 3×3 convolutions, respectively. Note that the last two convolutional networks are designed to have comparable FLOPs to AugLocal.

| | $(d = 2)$ | $(d = 3)$ | $(d = 4)$ | $(d = 5)$ | $(d = 6)$ | $(d = 7)$ | $(d = 8)$ | $(d = 9)$ |
|---|---|---|---|---|---|---|---|---|
| Unif. | 90.98±0.05 | 92.62±0.22 | 93.22±0.17 | 93.75±0.20 | 93.96±0.15 | 94.03±0.13 | 94.01±0.06 | 94.30±0.17 |
| Seq. | 82.19±0.47 | 85.34±0.18 | 86.88±0.23 | 88.52±0.06 | 89.49±0.12 | 90.44±0.17 | 91.15±0.17 | 91.85±0.04 |
| Repe. | 82.24±0.25 | 85.16±0.21 | 86.47±0.09 | 87.64±0.20 | 88.59±0.26 | 89.30±0.20 | 89.85±0.11 | 90.79±0.09 |
| C1×1 | 80.61±0.34 | 87.34±0.14 | 88.40±0.51 | 89.29±0.09 | 88.82±0.09 | 89.55±0.08 | 89.16±0.05 | 89.05±0.19 |
| C3×3 | 83.03±0.24 | 85.82±0.08 | 87.84±0.46 | 89.19±0.20 | 90.01±0.05 | 90.73±0.11 | 91.11±0.21 | 91.50±0.12 |

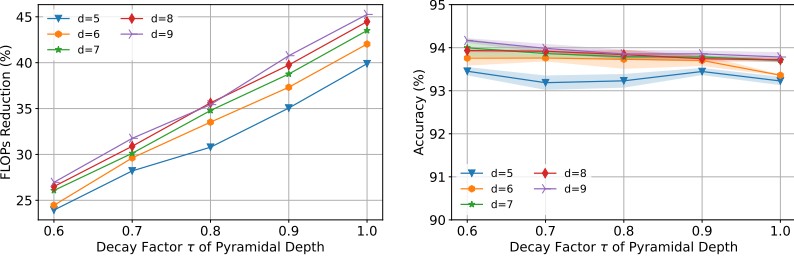

Figure 5: Influence of pyramidal depth on accuracy and computational efficiency. The FLOPs reduction is computed as the relative difference between with and without pyramidal depth. Refer to Table 11 for specific FLOPs values. Results are obtained using ResNet-110 on CIFAR-10.

## 4.4 MEMORY EFFICIENCY ANALYSIS

Local learning has the potential to improve memory efficiency compared to BP since intermediate variables such as activations are not necessarily kept in memory anymore after local updates (Nøkland & Eidnes, 2019). We provide empirical evidence of the memory efficiency on CIFAR-10 and ImageNet with ResNets. We report the minimally required memory that is aggregated across all GPUs for each task in Table 4. Results show that AugLocal can significantly reduce memory footprint compared to BP and gradient checkpoint (Chen et al., 2016) while maintaining comparable accuracies.

## 4.5 ABLATION STUDIES

We perform ablation studies to examine the effects of different auxiliary nets construction strategies and the pyramidal depth on CIFAR-10. Here, we choose ResNet-110 due to the large number of local layers it possesses, making it a suitable candidate for evaluation. Additionally, we increase the maximum depth limit of auxiliary networks to nine for a more comprehensive analysis.

**Comparison of different auxiliary network construction strategies** In AugLocal, we propose to build auxiliary networks by uniformly selecting a given number of layers from its subsequent layers in the primary network. To examine the effectiveness of this strategy, we compare it with another two: sequential (Seq.) and repetitive (Repe.) selection. Seq. uses the consecutive subsequent layers of a given hidden layer as its auxiliary network, while Repe. constructs the auxiliary network by repeating the same primary hidden layer to the desired depth. Results in Table 5 demonstrate that the uniform selection adopted in AugLocal consistently outperforms the others regarding accuracy. This highlights the effectiveness of AugLocal's auxiliary network construction approach in improving local learning. Note that the accuracy difference for $d = 2$ can be mainly attributed to the channel number difference. Both Seq. and Repe. tend to choose layers closer to the primary network layer, resulting in smaller channel numbers compared to the uniform one, leading to decreased accuracy.

To further demonstrate the role of our augmented auxiliary networks, we compare with handcrafted ones under comparable FLOPs. Specifically, we build auxiliary networks using commonly adopted 1×1 and 3×3 convolution layers, maintaining the same network depth while increasing the channel numbers. The results presented in Table 5 demonstrate a significant performance advantage of AugLocal over these primitive network construction methods, effectively validating the importance of augmented auxiliary networks. More ablation studies are provided in Appendix A.6.

**Influence of pyramidal depth** We empirically investigate the impact of pyramidal depth on model accuracy and computational efficiency. Specifically, with $d_{\min} = 2$ and $d$ varies from 5 to 9, we increase the decay factor of pyramidal depth from 0.6 to 1.0 with increments of 0.1. We then record both accuracy and FLOPs reduction. The FLOPs reduction is computed as the relative difference between the FLOPs of auxiliary networks with and without applying pyramidal depth. Figure 5 shows that increasing the decay factor leads to greater FLOPs reduction at the cost of slightly reduced accuracy. For example, by using the pyramidal depth method, we achieve a 45% reduction in FLOPs with a graceful accuracy degradation of only around 0.5%. This suggests the pyramidal depth method can effectively improve computational efficiency while maintaining high classification accuracy.

## 5 RELATED WORKS

**Local learning** was initially proposed as a pretraining method to improve global BP learning (Bengio et al., 2006; Hinton et al., 2006), which has soon gained attentions as an alternative to overcome limitations of BP. In addition to the supervised local learning rules discussed previously, there exist some other approaches. For instance, (Pyeon et al., 2021) proposes a differentiable search algorithm to decouple network blocks for block-wise learning and to select handcrafted auxiliary networks for each block. Some works employ self-supervised contrastive losses in their local learning rules (Illing et al., 2021), such as Contrastive Predictive Coding in GIM (Löwe et al., 2019), SimCLR in LoCo (Xiong et al., 2020), and Barlow Twins in (Siddiqui et al., 2023). LoCo additionally introduces coupling between adjacent blocks. However, most of these approaches are block-wise with only a few independent blocks, and it remains unclear whether they can effectively scale up to tens of local layers. A very recent work SoftHebb (Journé et al., 2023) explores local hebbian learning in soft winner-take-all networks, but its performance on ImageNet remains limited.

**Alternative learning rules to BP** have gained a growing interest in recent years (Lillicrap et al., 2020). Some efforts are devoted to addressing biologically unrealistic aspects of BP, such as the weight transport problem (Crick, 1989), by using distinct feedback connections (Lillicrap et al., 2016; Akrout et al., 2019), broadcasting global errors (Nø kland, 2016; Clark et al., 2021), random target projection (Frenkel et al., 2021), and activation sharing (Woo et al., 2021). Alternatively, target propagation (Le Cun, 1986; Bengio, 2014; Lee et al., 2015b) trains a distinct set of backward connections by propagating targets and using local reconstruction objectives. Baydin et al. (2022); Ren et al. (2023); Fournier et al. (2023) propose forward gradient learning to eliminate BP completely. To enable parallel training of layers, decoupled parallel BP (Huo et al., 2018b) and feature replay (Huo et al., 2018a) are proposed to update parameters with delayed gradients or activations. Decoupled neural interface (Jaderberg et al., 2017) uses synthetic gradients to tackle the update locking problem. These methods differ fundamentally from our method as they rely on global objectives. Most of them face challenges when scaling up to solve large-scale problems like ImageNet (Bartunov et al., 2018).

As a widely used regularization technique, stochastic depth (Huang et al., 2016) randomly drops a set of layers for each mini-batch, effectively mitigating the effect of layer coupling. In contrast, AugLocal uniformly samples some layers from the primary network to construct the auxiliary network, which happens before the training starts. These auxiliary networks are specifically used to optimize their respective hidden layers and do not have any direct impact on other layers in the primary network. Lee et al. (2015a) and Szegedy et al. (2015) utilize auxiliary networks to provide additional supervision for intermediate features in the primary network. However, the training of these auxiliary networks still affects their preceding layers as gradients are not decoupled across layers as in local learning.

## 6 CONCLUSION

We proposed a novel supervised local learning method named AugLocal that constructs auxiliary networks to enable hidden layers to be aware of their downstream layers during training. We evaluated AugLocal on various widely used large-scale network architectures and datasets including ImageNet. Our experimental results demonstrate that AugLocal consistently outperforms other supervised local learning methods by a large margin and achieves accuracy comparable to end-to-end BP. It is worth noting that a dedicated parallel implementation is required to take full advantage of the training efficiency of large-scale local learning rules, and we leave it as future work. In addition, to alleviate the update locking problem in auxiliary networks, one potential solution could involve gradually freezing some auxiliary layers during training, which will be investigated as part of our future work.

ACKNOWLEDGMENTS

This work was supported by the Research Grants Council of the Hong Kong SAR (Grant No. PolyU11211521, PolyU15218622, PolyU15215623, and PolyU25216423), The Hong Kong Polytechnic University (Project IDs: P0039734, P0035379, P0043563, and P0046094), and the National Natural Science Foundation of China (Grant No. U21A20512, and 62306259).

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

# A APPENDIX

## A.1 THEORETICAL ANALYSIS ON THE PARALLELIZATION OF AUGLOCAL

To analyze the parallelization, we compare the training time of AugLocal with that of BP. For simplicity, we assume the forward time $t_f$ and backward time $t_b$ of each layer to be the same. We denote the maximum depth of the auxiliary networks as $d$ and the depth of the primary network as $L + 1$, consistent with the notations in our paper. $N$ denotes the number of training iterations.

For BP training, the time to train $N$ iterations can be calculated as $(L + 1)(t_f + t_b)N$.

In AugLocal, we define a local layer as a hidden layer along with its associated auxiliary networks. The training time for any local layer $\ell$ per iteration can be represented as $(\ell-1)t_f+(d+1)(t_f+t_b)$. By parallelizing the training of these local layers once their inputs are available, the time of $(d+1)(t_f+t_b)$ can be shared among all local layers. Furthermore, starting from the second iteration, the forward pass of the $(\ell - 1)^{th}$ hidden layer can be parallelized with the backward pass of the $\ell^{th}$ auxiliary network. Based on these considerations, the training time of AugLocal for $N$ iterations can be calculated as $t_f L + (d + 1)(t_f + t_b)N$, which is approximated to $(d + 1)(t_f + t_b)N$ after omitting the constant term.

Consequently, the ratio of the training time between AugLocal and BP is approximately $\frac{d+1}{L+1}$. This suggests that as the maximum depth of the auxiliary network $d$ decreases, AugLocal demonstrates higher parallelization and faster training speed compared to BP. It is worth noting that to achieve the theoretical training speed-up, we need the customized parallel implementation that we consider as future work.

## A.2 IMPLEMENTATION DETAILS

Our experiments are based on four widely used benchmark datasets (i.e., CIFAR-10 (Krizhevsky et al., 2009), SVHN (Netzer et al., 2011), STL-10 (Coates et al., 2011), and ImageNet (Deng et al., 2009)). We compare our proposed AugLocal method with the end-to-end backpropagation (BP) (Rumelhart et al., 1985) algorithm and three state-of-the-art supervised local learning methods, including DGL (Belilovsky et al., 2020), PredSim (Nøkland & Eidnes, 2019), and InfoPro (Wang et al., 2021). We re-implement all of these methods in PyTorch using their official implementations [1]. We utilize consistent training configurations across all learning methods. All experiments are conducted on a machine equipped with $10\times$ NVIDIA RTX3090.

**Datasets** CIFAR-10 (Krizhevsky et al., 2009) dataset consists of 60K $32 \times 32$ colored images that are categorized into 10 classes with 50K images for training and 10K images for test. We use the standard data augmentation (He et al., 2016; Nøkland & Eidnes, 2019; Wang et al., 2021) in the training set, where 4 pixels are padded on each side of samples followed by a $32 \times 32$ crop and a random horizontal flip. SVHN (Netzer et al., 2011) dataset contains $32 \times 32$ digit images, each with a naturalistic background in RGB format. The standard split of 73,257 images for training and 26,032 images for test is adopted. Following Tarvainen & Valpola (2017); Wang et al. (2021), we augment training samples by padding 2 pixels on each side of images followed by a $32 \times 32$ crop. STL-10 (Coates et al., 2011) provides 5K labeled images for training and 8K labeled images for test. The size of each image is $96 \times 96$. Data augmentation is performed by $4 \times 4$ random translation followed by random horizontal flip (Wang et al., 2021). ImageNet (Deng et al., 2009) is a 1,000-class dataset with 1.2 million images for training and 50,000 images for validation. Following He et al. (2016); Huang et al. (2017); Wang et al. (2021), a $224 \times 224$ random crop followed by random horizontal flip is adopted for training samples, and a $224 \times 224$ resize and a central crop are applied for test samples.

**Training setups** For CIFAR-10, SVHN, and STL-10 experiments using ResNet-32 (He et al., 2016), ResNet-110 (He et al., 2016), and VGG19 (Simonyan & Zisserman, 2014), we use the SGD optimizer with a Nesterov momentum of 0.9 and the L2 weight decay factor of 1e-4. We adopt a batch size of 1024 on CIFAR-10 and SVHN and a batch size of 128 on STL10. We train the networks

---

[1]InfoPro: `https://github.com/blackfeather-wang/InfoPro-Pytorch`, PredSim: `https://github.com/anokland/local-loss`, and DGL: `https://github.com/eugenium/DGL`.

Table 6: Auxiliary networks in local learning methods for the three stages of ResNet-110. '/' is used to separate two auxiliary networks for two local losses in both PredSim and InfoPro. The former network is used for the cross-entropy loss, and the latter one serves another loss function.

| Stage | PredSim | InfoPro | DGL | AugLocal ($d = 2$) |
|---|---|---|---|---|
| 1 | AP-10FC / 16C3 | 32C3-AP-128FC-10FC / 12C3-3C3 | AP-16C1-16C1-16C1-AP-64FC-64FC-10FC | 64R-AP-10FC |
| 2 | AP-10FC / 32C3 | 64C3-AP-128FC-10FC / 12C3-3C3 | AP-32C1-32C1-32C1-AP-128FC-128FC-10FC | 64R-AP-10FC |
| 3 | AP-10FC / 64C3 | 64C3-AP-128FC-10FC / 12C3-3C3 | AP-64C1-64C1-64C1-AP-256FC-256FC-10FC | 64R-AP-10FC |

Table 7: Comparison of computational costs including FLOPs, GPU memory and computational overhead among PredSim, DGL, InfoPro and our proposed AugLocal method as well as BP and gradient checkpoint (Chen et al., 2016) on CIFAR-10 using the ResNet-110 architecture.

| | BP | Gradient Checkpoint | PredSim | DGL | InfoPro |
|---|---|---|---|---|---|
| FLOPs (G) | 0.25 | 0.25 | 0.25 | 0.26 | 0.34 |
| GPU Memory (GB) | 9.27 | 3.03 | 1.54 | 1.61 | 3.98 |
| Computational Overhead (Wall-clock Time) | - | 34.1% | 65.9% | 76.2% | 292.3% |
| Acc. | 94.61±0.18 | 94.61±0.18 | 74.95±0.36 | 85.69±0.32 | 86.95±0.46 |

| | AugLocal ($d = 2$) | AugLocal ($d = 3$) | AugLocal ($d = 4$) | AugLocal ($d = 5$) | AugLocal ($d = 6$) |
|---|---|---|---|---|---|
| FLOPs (G) | 0.63 | 0.69 | 0.80 | 0.98 | 1.13 |
| GPU Memory (GB) | 1.71 | 1.62 | 1.70 | 1.71 | 1.72 |
| Computational Overhead (Wall-clock Time) | 87.2% | 115.9% | 135.6% | 180.8% | 214.6% |
| Acc. | 90.98±0.05 | 92.62±0.22 | 93.22±0.17 | 93.75±0.20 | 93.96±0.15 |

for 400 epochs, setting the initial learning rate to 0.8 for CIFAR-10/SVHN and 0.1 for STL-10, with the cosine annealing scheduler (Loshchilov & Hutter, 2019). For ImageNet experiments, we train VGG13 (Simonyan & Zisserman, 2014) with an initial learning rate of 0.1 for 90 epochs, and train ResNet-34 (He et al., 2016) and ResNet-101 (He et al., 2016) with initial learning rates of 0.4 and 0.2 for 200 epochs, respectively. We set batch sizes of VGG13, ResNet-34, and ResNet-101 to 256, 1024, and 512, respectively. We keep other training configurations consistent with the ones on CIFAR-10. It is worth noting that, to reduce the computational costs of auxiliary networks, we change the number of hidden neurons in each auxiliary network's classifier from 4096 to 512 on VGG13.

## A.3 COMPARISON OF COMPUTATIONAL COSTS AMONG LOCAL LEARNING METHODS

**Auxiliary networks** We keep the original configurations as stated in their respective papers for the auxiliary networks of DGL, PredSim, and InfoPro in our experiments. The auxiliary networks in these local learning baselines and AngLocal ($d = 2$) are provided in Table 6. For clarity, we show the auxiliary nets based on the three stages of ResNet-110, each stage with the same number of output channels. We use the following notations: R denotes a residual block, C is a convolutional layer, AP signifies average pooling, and FC indicates a fully-connected layer. C1 and C3 refer to 1×1 and 3×3 convolutional kernel sizes, respectively. The value preceding C, R, and FC denotes the number of output channels.

**Computational costs** We compare the computational costs including FLOPs, GPU memory, and wall-clock time among BP, PredSim, DGL, InfoPro and our proposed AugLocal method. Note that the wall-clock time across all local learning methods is measured specifically under the sequential implementation setting, where each hidden layer is trained sequentially after receiving a batch of samples. Table 7 demonstrates that AugLocal achieves significantly higher accuracy at slightly higher computational costs than the other methods using the ResNet-110 architecture on CIFAR-10. The accuracy of AugLocal can be further improved by employing deeper auxiliary networks and larger computational costs. Additionally, we further compare AugLocal with gradient checkpointing (Chen et al., 2016). Our results in Table 7 demonstrate that AugLocal can achieve a much lower GPU memory footprint than gradient checkpoint, albeit accompanied by a moderate increase in wall-clock time. It is worth noting that the actual memory overhead of AugLocal does not follow a perfect linear

growth due to the PyTorch backward implementation has been optimized for e2e BP training. In our future work, we will explore more efficient CUDA implementation to address this issue.

## A.4 GENERALIZATION TO DIFFERENT CONVNETS

To evaluate the generalization ability of AugLocal across different convolutional networks (ConvNets), we conduct experiments on three popular ConvNets: MobileNet (Sandler et al., 2018), EfficientNet (Tan & Le, 2019), and RegNet (Radosavovic et al., 2020).

**Network architectures** MobileNetV2 (Sandler et al., 2018) is a lightweight architecture and comprises two types of building blocks. One is the inverted bottleneck residual block with a stride of 1, and another is the variant with a stride of 2 for downsizing. Each block contains 3 convolutional layers, including two point-wise convolution and one depth-wise convolution. EfficientNetB0 (Tan & Le, 2019) employs a compound scaling strategy to jointly scale network's depth, width, and resolution, offering a superior performance in terms of efficiency. The building block of EfficientNetB0 is the inverted residual block with an additional squeeze and excitation (SE) layer. RegNetX_400MF (Radosavovic et al., 2020) is derived from a low-dimensional network design space consisting of simple and regular networks. The standard residual bottleneck blocks with group convolution are adopted as its building blocks, each of which comprises a $1 \times 1$ convolution, followed by a $3 \times 3$ group convolution and a final $1 \times 1$ convolution.

**Training setups** As the minimal indivisible units, the building blocks in the three architectures are their local layers, which are independently trained with local learning rules. For AugLocal, the downsampling operation in auxiliary networks is performed by changing the stride of the corresponding auxiliary layer to 2. Other training configrations are the same as the previous ones on CIFAR-10.

Our experimental results in Table 8 demonstrate that AugLocal consistently obtains comparable accuracy to BP, regardless of the network structure, highlighting the potential of AugLocal to generalize across different network architectures.

Table 8: Performances of AugLocal on different ConvNets. The experiments are conducted on CIFAR-10.

|  | BP | AugLocal ($d = 3$) | ($d = 4$) | ($d = 5$) | ($d = 6$) |
|---|---|---|---|---|---|
| MobileNetV2 ($L = 19$) | 94.89±0.15 | 92.16±0.24 | 93.94±0.23 | 94.43±0.04 | **94.52±0.08** |
| EfficientNetB0 ($L = 17$) | 93.52±0.15 | 92.70±0.14 | 92.84±0.15 | 93.03±0.16 | **93.13±0.08** |
| RegNetX_400MF ($L = 23$) | 95.70±0.12 | 94.42±0.11 | 94.72±0.01 | 94.96±0.12 | **95.09±0.10** |

## A.5 COMPARISON OF LOCAL LEARNING METHODS WITH COMPARABLE FLOPS

This experiment compares AugLocal to other local learning methods with comparable FLOPs. We scale up the auxiliary networks of DGL by using 3×3 convolutions with the same network depth and a multiplier to scale up the channel numbers of the convolutional layers to ensure similar FLOPs as AugLocal. We further implement PredSim and InfoPro, which incorporate additional local losses and auxiliary networks, resulting in higher FLOPs than DGL. The ResNet-110 architecture on the CIFAR-10 dataset is adopted in this experiment. Our results in Table 9 consistently demonstrate that AugLocal outperforms the other methods with similar FLOPs, reaffirming the effectiveness of our approach in constructing auxiliary networks for improved performance in supervised local learning.

## A.6 ABLATION STUDY OF AUGLOCAL'S AUXILIARY NETWORKS

We conduct ablation experiments to investigate the impact of altering the auxiliary architecture on AugLocal's performance. In this ablation study, we focus on AugLocal with a depth ($d$) of 6, which equals the depth of DGL Belilovsky et al. (2020). We use the ResNet-110 architecture that has three stages, each having the same number of output channels. To align with DGL, which uses the same auxiliary networks for layers in the same stage, we gradually modify AugLocal's auxiliary architectures to match those of DGL.

Initially, we replace the auxiliary networks in the second stage with a repetitive selection strategy combined with downsampling while keeping the other two stages unchanged. This modification

Table 9: Comparison of local learning methods with comparable FLOPs on ResNet-110.

| Method | $(d = 2)$ | $(d = 3)$ | $(d = 4)$ | $(d = 5)$ | $(d = 6)$ |
|---|---|---|---|---|---|
| AugLocal | 90.98±0.05 | 92.62±0.22 | 93.22±0.17 | 93.75±0.20 | 93.96±0.15 |
| DGL | 83.03±0.24 | 85.82±0.08 | 87.84±0.46 | 89.19±0.20 | 90.01±0.05 |
| PredSim | 72.06±0.63 | 80.16±0.47 | 86.00±0.56 | 88.27±0.72 | 88.34±0.34 |
| InfoPro | 83.71±0.20 | 89.14±0.17 | 90.75±0.22 | 91.45±0.05 | 92.10±0.17 |

results in a 1.21% accuracy drop. Subsequently, we remove the downsampling operation, leading to a further accuracy drop of 1.02%. Based on these modified auxiliary architectures, we replace the first stage layers with the repetitive auxiliary networks and downsampling, resulting in an accuracy degradation of around 1.5%. Finally, we replace all residual blocks in the auxiliary networks with $3 \times 3$ convolutional layers while maintaining the same number of channels. This change significantly affects the accuracy, resulting in a drop to 88.28%. It is worth noting that DGL further adopts convolutional $1 \times 1$ layers and fully-connected layers, which achieve a baseline accuracy of 85.69%.

These ablation experiments clearly demonstrate the important role of each component in the auxiliary networks of AugLocal. The results highlight the effectiveness of our approach in constructing auxiliary networks for improved performance in local learning.

We provide the details of auxiliary networks in the ablation experiment in Table 10. For consistency, we adopt the same notations as A.3, with the addition of s2 representing a stride of 2 for downsampling.

Table 10: Results of the ablation study for AugLocal by gradually modifying auxiliary networks to match those of the baseline.

| Method | Acc. | Stage 1 | Stage 2 | Stage 3 |
|---|---|---|---|---|
| AugLocal $(d = 6)$ | 93.96±0.15 | Uniform Selection | Uniform Selection | Uniform Selection |
| Replace with repe. and downsampling (ds.) in Stage 2 | 92.75±0.09 | Uniform Selection | 32Rs2-32R-32R-32R-32R-AP-10FC | Uniform Selection |
| Replace with repe. in Stage 2 | 91.73±0.11 | Uniform Selection | 32R-32R-32R-32R-32R-AP-10FC | Uniform Selection |
| Replace with repe. and ds. in both Stage 1 and 2 | 91.40±0.08 | 16Rs2-16R-16R-16R-16R-AP-10FC | 32Rs2-32R-32R-32R-32R-AP-10FC | Uniform Selection |
| Replace with repe. and ds. in Stage 1 and with repe. in Stage 2 | 89.92±0.37 | 16Rs2-16R-16R-16R-16R-AP-10FC | 32R-32R-32R-32R-32R-AP-10FC | Uniform Selection |
| Replace with 3×3 convlutional layers and downsampling in all stages | 88.28±0.24 | AP-16C3-16C3-16C3-16C3-16C3-AP-10FC | AP-32C3-32C3-32C3-32C3-32C3-AP-10FC | AP-64C3-64C3-64C3-64C3-64C3-AP-10FC |
| DGL | 85.69±0.32 | AP-16C1-16C1-16C1-AP-64FC-64FC-10FC | AP-32C1-32C1-32C1-AP-128FC-128FC-10FC | AP-64C1-64C1-64C1-AP-256FC-256FC-10FC |

## A.7 RESULTS OF AUGLOCAL ON DOWNSTREAM TASKS

To evaluate the generalization ability of AugLocal on downstream tasks, we conduct experiments on the challenging COCO dataset (Lin et al., 2014) for object detection and instance segmentation. Following the common practice, we use the pre-trained ResNet-34 on ImageNet as a backbone and integrate it with the Mask R-CNN detector (He et al., 2017). To ensure fair comparisons, we maintain consistent training configurations for both AugLocal and BP. Specifically, we utilize the AdamW optimizer, a $1\times$ training schedule consisting of 12 epochs and a batch size of 16. The results in Table 12 show that AugLocal consistently achieves comparable performance to BP across all average precision (AP) metrics, suggesting that AugLocal can effectively generalize pre-trained models for downstream tasks.

## A.8 CONVERGENCE SPEED

In this experiment, we aim to investigate the convergence speed of our proposed AugLocal method by comparing it with BP and other local learning rules. To this end, we visualize the learning curves of these methods on ResNet-32 and ResNet-110. As shown in Figure 6, AugLocal achieves a faster decrease in the network output loss as compared to other local learning rules. Moreover, as the

Table 11: Influence of pyramidal depth on computational efficiency. This complements Figure 5 with explicit values of FLOPs (G). The FLOPs for BP is 0.25G.

|       | $\tau = 1$ | $\tau = 0.9$ | $\tau = 0.8$ | $\tau = 0.7$ | $\tau = 0.6$ |
|-------|------------|--------------|--------------|--------------|--------------|
| $d = 5$ | 0.79 | 0.83 | 0.87 | 0.89 | 0.93 |
| $d = 6$ | 0.90 | 0.95 | 0.99 | 1.03 | 1.09 |
| $d = 7$ | 0.99 | 1.06 | 1.11 | 1.17 | 1.22 |
| $d = 8$ | 1.09 | 1.16 | 1.22 | 1.29 | 1.36 |
| $d = 9$ | 1.18 | 1.26 | 1.35 | 1.41 | 1.49 |

Table 12: Results of AugLocal on the COCO object detection and instance segmentation benchmarks.

| Method | $AP^b$ | $AP^b_{50}$ | $AP^b_{75}$ | $AP^m$ | $AP^m_{50}$ | $AP^m_{75}$ |
|--------|--------|-------------|-------------|--------|-------------|-------------|
| BP | 36.3 | 56.4 | 39.4 | 33.8 | 53.9 | 36.1 |
| AugLocal | 36.2 | 56.0 | 39.1 | 33.4 | 53.2 | 35.7 |

Table 13: Performance of AugLocal with the InfoPro loss (Wang et al., 2021) on ResNet-110. The results of AugLocal with the cross-entropy (CE) loss are provided as a baseline.

| Loss | $(d = 2)$ | $(d = 3)$ | $(d = 4)$ | $(d = 5)$ | $(d = 6)$ | $(d = 7)$ | $(d = 8)$ | $(d = 9)$ |
|------|-----------|-----------|-----------|-----------|-----------|-----------|-----------|-----------|
| CE | 90.98±0.05 | 92.62±0.22 | 93.22±0.17 | 93.75±0.20 | 93.96±0.15 | 94.03±0.13 | 94.01±0.06 | 94.30±0.17 |
| InfoPro | 91.59±0.11 | 92.75±0.50 | 93.71±0.14 | 94.11±0.19 | 94.02±0.09 | 94.17±0.13 | 94.29±0.07 | 94.08±0.18 |

depth of auxiliary networks increases, the convergence speed of AugLocal improves and gradually approaches the one of BP. This finding reconfirms the efficacy of our AugLocal method in optimizing networks to achieve high performance.

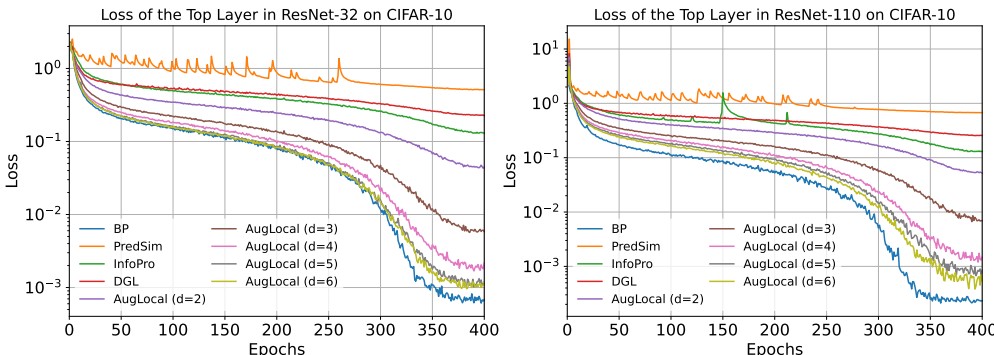

Figure 6: Learning curves of different learning rules on ResNet-32 and ResNet-110. The y-axis is in log scale.

## A.9 SYNERGY BETWEEN AUGLOCAL AND INFOPRO

Our proposed AugLocal method is orthogonal to existing supervised local learning works (Nøkland & Eidnes, 2019; Wang et al., 2021) that propose advanced local loss functions. In this part, we investigate the potential benefits of combining AugLocal with InfoPro (Wang et al., 2021). Specifically, each hidden layer additionally incorporates a reconstruction loss with an auxiliary network. Following (Wang et al., 2021), we adopt the auxiliary network containing two convolutions and up-sampling operations. It is worth noting that the original augmented auxiliary network with the cross-entropy loss in each hidden layer keeps unchanged.

The results in Table 13 demonstrate that incorporating the additional reconstruction loss can lead to accuracy improvements in most cases. This suggests that AugLocal can generalize and synergize with advanced local objectives to improve performance further.

