# OpenReview forum: "Scaling Supervised Local Learning with Augmented Auxiliary Networks"
_ICLR.cc/2024/Conference — ICLR 2024 poster_

### Official Review · Reviewer_ptT3 · 2023-10-30

**Soundness:** 3 good
**Presentation:** 3 good
**Contribution:** 3 good
**Rating:** 6
**Confidence:** 4

**Summary:**

This paper attempts to use local learning which update each layer of the neural networks in an isolated way to reduce the huge memory consumption of classical contrastive learning. As mentioned by the authors, such kind of update neglects the dependence of different layers of neural networks which leads to significant performance drop. To alleviate this problem, the authors propose to select a small subset of layers and train them together during local learning. For different layer, the algorithm takes different number of layers to construct the subset. To validate the performance of the proposed method, the authors conduct extensive experiments on several widely used datasets such as CIFAR-10, SVHN, STL-10 and ImageNet. The results illustrate that the performance can be decent with reduction of 40% GPU memory.

**Strengths:**

The intuition of the proposed method is reasonable, taking several layers to do local learning should help enhance the performance of local learning via importing information from other layers.
The detailed method is direct, select layers by some step from the current layer to the output layer.
The performance on several small-scale dataset is decent, deep networks such as ResNet-110 can be trained using the proposed method with small performance gap compared with BP. The GPU memory reduction is very significant.
The authors also validate the performance on different CNN network structures.
On several CNN network structures as well as several small-scale dataset, the proposed method outperform the counterparts by a significant margin.
The authors also analyze the method with extensive ablation studies.

**Weaknesses:**

The performance gap between AugLocal and BP seems to be enlarged on ImageNet. The authors also do not show the experiments on vision transformer which is also a kind of widely used structure in computer vision.

Could the authors present the performance comparison when using the model trained using BP and Auglocal for downstream task fine-tuning?

**Questions:**

Please refer to weakness.

---

> ### Author Response · Authors · 2023-11-17
>
> Thank you for taking the time to review our work and providing us with positive and helpful comments and suggestions. In response to your raised questions, we have provided explanations below for both of them.
>
> **Q1:** The performance gap between AugLocal and BP seems to be enlarged on ImageNet. The authors also do not show the experiments on vision transformer which is also a kind of widely used structure in computer vision.
>
> **A1:** Thank you for your comments. Regarding your first comment, we would like to clarify that most of the existing supervised local learning rules have not been extensively evaluated on ImageNet using the challenging layer-wise local learning setting. Our AugLocal stands out as the first approach to scale up to tens of local layers while achieving comparable accuracy to BP on ImageNet.
>
> Regarding the absence of experiments on vision transformers, we believe it is an interesting topic to explore independently as the local learning research on vision transformers is not yet available. It would require more time to build the baselines for a comprehensive comparison and, therefore, we would like to leave it as future works. Nevertheless, we would like to highlight that we have performed extensive experiments on numerous convolutional neural architectures to demonstrate the promising generalization capability of the proposed AugLocal rule.
>
> **Q2:** Could the authors present the performance comparison when using the model trained using BP and Auglocal for downstream task fine-tuning?
>
> **A2:** Thank you very much for your suggestion. We are currently in the process of conducting experiments on downstream tasks (object detection and segmentation), and we will update the results in our final paper after the experiments are completed.

---

> ### Author Response · Authors · 2023-11-21
> **Results of AugLocal on Downstream Tasks**
>
> We have completed the experiments on the COCO dataset for object detection and instance segmentation to evaluate the generalization ability of AugLocal on downstream tasks. Following the common practice, we use the pre-trained ResNet-34 on ImageNet as a backbone and integrate it with the Mask R-CNN detector. To ensure fair comparisons, we maintain consistent training configurations for both AugLocal and BP. Specifically, we utilize the AdamW optimizer, a 1x training schedule consisting of 12 epochs and a batch size of 16.
>
> The table below shows that AugLocal consistently achieves comparable performance to BP across all average precision (AP) metrics. This suggests that AugLocal can effectively generalize pre-trained models for downstream tasks.
>
> | Method   | $AP^b$ | $AP_{50}^b$ | $AP_{75}^b$ | $AP^m$ | $AP_{50}^m$ | $AP_{75}^m$ |
> |----------|--------|-------------|-------------|--------|-------------|-------------|
> | BP       | 36.3   | 56.4        | 39.4        | 33.8   | 53.9        | 36.1        |
> | AugLocal | 36.2   | 56.0        | 39.1        | 33.4   | 53.2        | 35.7        |
>
> We have incorporated these experiments into Appendix A.7 of our paper. We hope that these updated results adequately address your concern regarding the effectiveness of AugLocal for downstream tasks fine-tuning.

---

> > ### Comment · Reviewer_ptT3 · 2023-11-23
> >
> > I appreciate the response, I will keep my score.

---

### Official Review · Reviewer_6ww4 · 2023-10-30

**Soundness:** 3 good
**Presentation:** 3 good
**Contribution:** 3 good
**Rating:** 8
**Confidence:** 5

**Summary:**

Unlike traditional local learning techniques, AugLocal does not create a new auxiliary network. Instead, it is constructed by selecting a few layers from the subsequent layers of the hidden layers. With this approach, AugLocal was able to achieve higher accuracy compared to traditional local learning, and it is comparable to backpropagation.

**Strengths:**

1. This paper is well-written and easy to understand.

2. Unlike traditional methods, the concept of utilizing existing hidden layers to construct auxiliary networks, thereby learning representations related to the global loss, is novel.

3. The representation similarity analysis convincingly shows that AugLocal learns in a manner more akin to backpropagation compared to traditional local learning.

4. Not only did it achieve higher accuracy compared to traditional local learning, but it also saved GPU memory.

**Weaknesses:**

1. Fair comparison to previous works
: The author presents comparison results between AugLocal, DGL, and InfoPro. However, looking at Table 2 in the Appendix, AugLocal has higher FLOPs than the other methods. Typically, networks with a larger number of FLOPs tend to achieve higher accuracy. Therefore, it seems appropriate to compare the accuracy among AugLocal, DGL, and InfoPro with the same FLOPs, that is, auxiliary networks with equivalent FLOPs.

2. Do not solve update locking perfectly
: AugLocal utilizes the subsequent layers of the hidden layers. As a result, this method has to wait for the subsequent layers to be trained for training the current layer with the next data (update locking). In contrast, traditional local learning methods like DGL and InfoPro completely solve this update locking issue, enabling asynchronous training.

3. (Minor) Another tasks like GNNs or NLP
: The author demonstrated the efficacy of AugLocal in image classification tasks using models like ResNet, EfficientNet, and MobileNetv2 for the sake of generality. If AugLocal also performs well in graph tasks or NLP, it would further attest to its generality.

**Questions:**

Despite the aforementioned shortcomings, I find the idea of using the subsequent layers of the hidden layer as an auxiliary network to be very innovative. As a result, I have awarded a score of 6. However, I believe addressing the following questions could further elevate the quality of the paper.

1. Comparisons in same FLOPs
 : I acknowledge that it might be challenging to compare with AugLocal at the same FLOPs since the method of determining the auxiliary network in DGL and InfoPro is already fixed. Nevertheless, if there is a comparison result with AugLocal at the same FLOPs, it would enhance the quality of the paper if it is possible.

2. Refer the update locking issue as limitation
: As mentioned above, due to the nature of the algorithm, the current layer cannot be trained with the next data until the subsequent layer of the hidden layer is trained (update locking). It would be good to mention this limitations in the paper's conclusion or Appendix and suggest directions for future work. If possible, briefly mentioning a possible solution would also be beneficial.

3. Recent related works
: The author discusses algorithms like FA (2016), DFA(2016), and Weight-Mirror (2019) in section 5 to address the weight transport issue. However, there are more recent algorithms that have proposed. Please add and compare AugLocal with DRTP (2021) and ASAP (2021), which are recent biologically plausible learning rules to solve weight transport problem, to section 5.

(1) Learning Without Feedback: Fixed Random Learning Signals Allow for Feedforward Training of Deep Neural Networks, 2021

(2) Activation Sharing with Asymmetric Paths Solves Weight Transport Problem without Bidirectional Connection, 2021

4. (minor) Perform another tasks
 : It would be even better if results for simple models like GCN2 or BERT-Base could be added.

If requests 1-3 are perfectly addressed, I am willing to raise the score to 7-8 points.

**# After the discussion, all my concerns have been resolved, and I will raise the score to 8.**

---

> ### Author Response · Authors · 2023-11-17
>
> Thank you for your positive recognition on our work as well as your helpful comments. We would like to address all of your requests in the following.
>
> **Q1:** Comparisons in same FLOPs : I acknowledge that it might be challenging to compare with AugLocal at the same FLOPs since the method of determining the auxiliary network in DGL and InfoPro is already fixed. Nevertheless, if there is a comparison result with AugLocal at the same FLOPs, it would enhance the quality of the paper if it is possible.
>
> **A1:** Thank you for your suggestion. In response to your comment, we have performed additional experiments to scale up the auxiliary networks of DGL by using 3x3 convolutions with the same auxiliary network depth and a multiplier to scale up the channel numbers of the convolutional layers to ensure similar FLOPs as AugLocal. We further implement PredSim and InfoPro, which incorporate additional local losses and auxiliary networks, resulting in higher FLOPs than DGL. Our results in the table below consistently demonstrate that AugLocal outperforms the other methods with similar FLOPs. We have incorporated this experiment in Appendix A.5 of the updated paper.
>
> | Method | d=2 | d=3 | d=4 | d=5 | d=6 |
> |:---:|:---:|:---:|:---:|:---:|:---:|
> | AugLocal | 90.98±0.05 | 92.62±0.22 | 93.22±0.17 | 93.75±0.20 | 93.96±0.15 |
> | DGL | 83.03±0.24 | 85.82±0.08 | 87.84±0.46 | 89.19±0.20 | 90.01±0.05 |
> | PredSim | 72.06±0.63 | 80.16±0.47 | 86.00±0.56 | 88.27±0.72 | 88.34±0.34 |
> | InfoPro | 83.71±0.20 | 89.14±0.17 | 90.75±0.22 | 91.45±0.05 | 92.10±0.17 |
>
> **Q2:**	Refer the update locking issue as limitation : As mentioned above, due to the nature of the algorithm, the current layer cannot be trained with the next data until the subsequent layer of the hidden layer is trained (update locking). It would be good to mention this limitations in the paper's conclusion or Appendix and suggest directions for future work. If possible, briefly mentioning a possible solution would also be beneficial.
>
> **A2:** We would like to clarify that the construction of augmented auxiliary networks happens before the training starts, and the selected auxiliary layers only share the same structures with their counterparts in the primary network. This particular design ensures hidden layers in the primary network can be updated independently and asynchronously, thereby solving the update locking problem. However, as already mentioned in the paper, the update locking problem still exists in the training of the hidden layers along with their auxiliary networks and will become exacerbated when the depth of auxiliary networks grows. Therefore, we constrain the depth of auxiliary networks to control the degree of update locking. To alleviate the update locking problem in auxiliary networks, one potential solution could involve gradually freezing some auxiliary layers during training. This strategy will be investigated as part of our future work. Based on your suggestions, we have updated to mention this point in the paper's conclusion. Thanks.
>
> **Q3:** Recent related works : The author discusses algorithms like FA (2016), DFA(2016), and Weight-Mirror (2019) in section 5 to address the weight transport issue. However, there are more recent algorithms that have proposed. Please add and compare AugLocal with DRTP (2021) and ASAP (2021), which are recent biologically plausible learning rules to solve weight transport problem, to section 5.
>
> **A3:** Thank you for sharing these excellent references. We have incorporated both of them into Section 5 of our revised paper.
>
> **Q4:** Perform another tasks : It would be even better if results for simple models like GCN2 or BERT-Base could be added.
>
> **A4:** Thank you for raising the potential applicability of our proposed method on graph neural networks and the transformer architecture model. We believe they are interesting topics to explore independently as the local learning research on these architectures is not yet available. It would require more time to build the baselines for a comprehensive comparison and, therefore, we would like to leave it as future works.

---

> > ### Comment · Reviewer_6ww4 · 2023-11-23
> >
> > I am grateful for the author's thorough response. Particularly, I find it meaningful that AugLocal achieves higher accuracy with the same FLOPs. Although the weight update issue is not entirely resolved, the approach of utilizing subsequent layers as auxiliary layers is innovative and, in my opinion, needs further exploration in future work. Therefore, I have decided to increase my rating to 8.

---

### Official Review · Reviewer_SLqf · 2023-10-31

**Soundness:** 2 fair
**Presentation:** 3 good
**Contribution:** 2 fair
**Rating:** 5
**Confidence:** 3

**Summary:**

This paper studies the supervised local learning problem. It proposes a new rule to construct auxiliary networks for each local layer/block: using a subset of the following layers and the classifier as its auxiliary net, thus closing the gap between local learning and end-to-end learning. Experiments are conducted on CIFAR-10, SVHN, STL-10 and ImageNet. The experimental results show that constructing the auxiliary networks in such a way can effectively close the gap between local learning and bp-based end-to-end learning with tens of local layers/blocks while keeping the memory footprint low.

**Strengths:**

1. The overall idea is straightforward and intuitive.
2. The performance looks impressive.
3. The representation similarity looks interesting to me.

**Weaknesses:**

I have several questions regarding the experimental results and design details in the Question section. Unfortunately, given the current version of the paper, it is not clear why the proposed method works well, what plays the critical role (downsampling in the auxiliary network? shared classifier?) and what is the overhead of the proposed updating rule. I'd be happy to raise my rating if the authors could address those questions during the discussion period, for now, my rating would be 3.

**Questions:**

1. It is not clear when the authors say: "when more layers of the primary network are selected into the auxiliary networks". Does that mean the weights are shared between layers in the primary network and the auxiliary network? Or just the initialization is shared, or just the architecture is shared?

2. Based on 1, My guess is that other layers are not shared based on the last paragraph of Sec 3.2 and the last sentence in A.2 in the supplementary. In this case, it is not clear why Unif., Seq., Repe. can show a large difference even for d = 2 in Table 5. Does that mean downsampling in the auxiliary network is critical? If the authors want to claim the importance of architectural bias, I believe the missing results are C1×1/C3x3 with downsampling, which should not be considered architectural bias from my perspective. The authors should also show C3x3 performs equally well with the VGG network. The authors should also ablate to what circumstances, when we alter the auxiliary architecture presented in Table 1 in the supplementary, the AugLocal performance degraded and performed similarly to other baselines, simply attributing the performance gain to architecture bias is not convincing as the learning is free-form if the auxiliary network weights are not shared with the primary network.

3. Table 4 shows weird results; how did the authors get the 157.12 GB GPU memory results as there is no single GPU that can handle it AFAIK? The ResNet-110 result looks like an outlier. Can the authors elaborate on why it shows significantly better memory saving while the architecture is similar to other cases (ResNet-32/34/101)?

4. It would be great to show FLOPs comparison between local learning and end-to-end learning instead of just showing FLOPs reduction in Figure 5. Also, gamma is introduced in eq2 but never used in the following text, making it confusing about how it guides the rule design.

---

> ### Author Response · Authors · 2023-11-17
> **Response to Reviewer SLqf (1/3)**
>
> Thank you for your constructive comments and suggestions.  In the following, we provide point-to-point replies to all your comments. We sincerely hope our efforts help to address your concerns.
>
> **Q1:** It is not clear when the authors say: "when more layers of the primary network are selected into the auxiliary networks". Does that mean the weights are shared between layers in the primary network and the auxiliary network? Or just the initialization is shared, or just the architecture is shared?
>
> **A1:** Thank you for bringing this up. We would like to clarify that only the architecture is shared in our method. To improve the clarity on this point, we have revised Section 3.2 of our paper, emphasizing that "the selection of the auxiliary network takes place prior to training, and only the architectural structure of the selected layers are employed in the construction of the auxiliary network.".
>
> **Q2.1:**	It is not clear why Unif., Seq., Repe. can show a large difference even for d = 2 in Table 5.
>
> **A2.1:** The large accuracy difference for d=2 can be attributed to the different channel numbers used in their residual blocks. The sequential (Seq.) and repetitive (Repe.) selection strategies will select layers that are closer to the primary network layer that we intend to train, which have smaller channel numbers compared to those chosen by the uniform (Unif.) sampling method. This leads to a noticeable decrease in accuracy. We have revised  Section 4.5 of our paper to provide a detailed explanation on this point.
>
> **Q2.2:**	If the authors want to claim the importance of architectural bias, I believe the missing results are C1×1/C3x3 with downsampling, which should not be considered architectural bias from my perspective. The authors should also show C3x3 performs equally well with the VGG network.
>
> **A2.2:** Our experiments involving C1×1/C3×3 are specifically designed to demonstrate the advantage of our proposed AugLocal strategy compared to other handcrafted ways to construct auxiliary networks. Following your suggestion, we conduct an experiment of C3×3 with downsampling at comparable FLOPs to AugLocal. Our experimental results in the table below demonstrate improved accuracies compared to those models without downsampling. Furthermore, it is evident that AugLocal consistently outperforms C3×3 with downsampling, suggesting the effectiveness of the proposed auxiliary network construction strategy in AugLocal (i.e., perform uniform sampling from the subsequent layers in the primary network).
>
> For the VGG networks used in this paper, our augmented auxiliary networks were constructed exactly following the way you have suggested (i.e., uniformly sample the C3×3 layers from the subsequent layers in the primary network). This approach delivers superior performance for VGG networks, as illustrated by the results presented in Table 2 of our paper.
>
> | |d=2|d=3|d=4|d=5|d=6|
> |:-:|:-:|:-:|:-:|:-:|:-:|
> |AugLocal|90.98±0.05|92.62±0.22|93.22±0.17|93.75±0.20|93.96±0.15|
> |C3×3 **without** downsampling|83.03±0.24|85.82±0.08|87.84±0.46|89.19±0.20|90.01±0.05|
> |C3×3 **with** downsampling|85.18±0.26 |89.90±0.13|91.36±0.23|91.72±0.11|91.97±0.18|

---

> ### Author Response · Authors · 2023-11-17
> **Response to Reviewer SLqf (2/3)**
>
> **Q2.3:**	The authors should also ablate to what circumstances, when we alter the auxiliary architecture presented in Table 1 in the supplementary, the AugLocal performance degraded and performed similarly to other baselines, simply attributing the performance gain to architecture bias is not convincing as the learning is free-form if the auxiliary network weights are not shared with the primary network.
>
> **A2.3:**	 Thank you very much for your suggestion, which we believe is very helpful to understand the architecture bias in our proposed AugLocal learning. Following your suggestion, we have conducted ablation studies to investigate the impact of altering the auxiliary network architecture on AugLocal's performance. Particularly, we use the ResNet-110 architecture with three stages, each having the same number of output channels. To align with DGL, which uses the same auxiliary networks for all layers within the same stage, we gradually modify AugLocal's auxiliary architectures to match those of DGL.
>
> Initially, we replace the auxiliary networks in the second stage with a repetitive selection strategy combined with downsampling while keeping the other two stages unchanged. This modification results in a 1.21% accuracy drop. Subsequently, we remove the downsampling operation, leading to a further accuracy drop of 1.02%. Based on both modified auxiliary architectures, we further replace the first stage layers with the repetitive auxiliary networks and downsampling, resulting in an accuracy degradation of around 1.5%. Finally, we replace all residual blocks in the auxiliary networks with 3x3 convolutional layers while maintaining the same number of channels. This change significantly affects the accuracy, resulting in a drop to 88.28%. It is worth noting that DGL further adopts convolutional 1x1 layers and fully-connected layers, which achieve a baseline accuracy of 85.69%. These ablation experiments clearly demonstrate the important role of each component in the auxiliary networks of AugLocal. The results highlight the effectiveness of our approach in constructing auxiliary networks for improved performance in local learning.
>
> We provide the details of auxiliary networks in the ablation experiment in the table below. For clarity, we use the following notations: R denotes a residual block, C is a convolutional layer, AP signifies average pooling, and FC indicates a fully-connected layer. C1 and C3 refer to 1x1 and 3x3 convolutional kernel sizes, respectively. The value preceding C, R, and FC denotes the number of output channels. s2 represents a stride of 2 for downsampling.
>
> We have updated our paper to include all of these ablation experiments in Appendix A.6.
>
> | Method | Acc.  | Stage 1 | Stage 2 | Stage 3 |
> |:---:|:---:|:---:|:---:|:---:|
> | AugLocal(d=6) | 93.96±0.15 | Uniform Selection | Uniform Selection | Uniform Selection |
> | Replace with repe. and  downsampling (ds.) in Stage 2 | 92.75±0.09 | Uniform Selection | 32Rs2-32R-32R -32R-32R-AP-10FC | Uniform Selection |
> | Replace with repe. in Stage 2 | 91.73±0.11 | Uniform Selection | 32R-32R-32R -32R-32R-AP-10FC | Uniform Selection |
> | Replace with repe. and  ds. in both Stage 1 and 2 | 91.40±0.08 | 16Rs2-16R-16R -16R-16R-AP-10FC | 32Rs2-32R-32R -32R-32R-AP-10FC | Uniform Selection |
> | Replace with repe. and ds.  in Stage 1 and with repe. in Stage 2 | 89.92±0.37 | 16Rs2-16R-16R -16R-16R-AP-10FC | 32R-32R-32R -32R-32R-AP-10FC | Uniform Selection |
> | Replace with 3x3 convlutional layers  and downsampling in all stages | 88.28±0.24 | AP-16C3-16C3-16C3 -16C3-16C3-AP-10FC | AP-32C3-32C3-32C3 -32C3-32C3-AP-10FC | AP-64C3-64C3-64C3 -64C3-64C3-AP-10FC |
> | DGL | 85.69±0.32 | AP-16C1-16C1-16C1- AP-64FC-64FC-10FC | AP-32C1-32C1-32C1- AP-128FC-128FC-10FC | AP-64C1-64C1-64C1- AP-256FC-256FC-10FC |

---

> ### Author Response · Authors · 2023-11-17
> **Response to Reviewer SLqf (3/3)**
>
> **Q3:** Table 4 shows weird results; how did the authors get the 157.12 GB GPU memory results as there is no single GPU that can handle it AFAIK? The ResNet-110 result looks like an outlier. Can the authors elaborate on why it shows significantly better memory saving while the architecture is similar to other cases (ResNet-32/34/101)?
>
> **A3:** Thank you for the feedback. First of all, we want to clarify that the reported GPU memory usage represents the aggregate utilization across all GPUs within the experimental machine, rather than the capacity of a single GPU card. This has been explicitly detailed in Appendix A.2 of our paper. Specifically, our experiments were conducted on a system equipped with 10 $\times$ NVIDIA RTX 3090 GPU cards with 24GB GPU memory each, meeting the memory requirements for our evaluations. To avoid any confusion, we have revised Section 4.4 of our paper to explicitly state this point for clarity.
>
> Regarding the observed memory efficiency of ResNet-110, it is essential to consider the underlying mechanics. In AugLocal, the memory allocated to a local layer (defined as a hidden layer along with its associated auxiliary network) is freed upon the completion of the local layer's update. As a result, the peak GPU memory consumption is determined by the local layer with the largest memory footprint, relative to other local layers. Therefore, the memory-saving characteristic of AugLocal stems from the ratio between the maximum memory utilization among local layers and the overall memory footprint of the entire network. As compared with other network architectures, ResNet-110 has a much smaller memory footprint of its local layers when contrasted with the total memory consumption of the network. Therefore, ResNet-110 shows substantial GPU memory saving in comparison to other network architectures.
>
> **Q4:** It would be great to show FLOPs comparison between local learning and end-to-end learning instead of just showing FLOPs reduction in Figure 5. Also, gamma is introduced in eq2 but never used in the following text, making it confusing about how it guides the rule design.
>
> **A4:** Thanks for your suggestions. We have updated our paper to provide explicit values of FLOPs for local learning and end-to-end learning in Appendix Table 6. In addition, the value of gamma is set to 0.5 in our experiments and we analyzed the influence of gamma in Figure 5. We have updated our paper to make this point clearer.
>
>
> **Q5:** What is the overhead of the proposed updating rule.
>
> **A5:** Despite adding additional auxiliary networks, AugLocal can achieve accelerated training speeds under a dedicated parallel implementation. As detailed in Appendix A.1, the theoretical training time ratio between AugLocal and BP can be approximated by  $\frac{d+1}{L+1}$, where $d$ and $L$ represent the depths of the auxiliary and primary networks, respectively. However, further efforts are required to develop the parallel implementation, which we have already included in the original paper's conclusion. Additionally, for completeness, we have provided the computational overhead of our method in Appendix Table 2 under the sequential implementation, where each hidden layer is trained sequentially after receiving a batch of samples. Our results demonstrate that AugLocal achieves higher accuracy at the expense of a slightly increased computational overhead compared to other local learning rules. Moreover, the accuracy of AugLocal can be further improved by leveraging deeper auxiliary networks, albeit with an associated increase in computational costs.

---

> ### Author Response · Authors · 2023-11-21
> **Kindly request your feedback on our responses**
>
> Dear Reviewer SLqf,
>
> As the deadline for the discussion period is approaching, we would like to kindly request your feedback on our responses.
>
> We wish to express our deepest gratitude for the time and efforts you have dedicated to reviewing our work. We sincerely hope that our detailed responses have adequately addressed all the concerns and suggestions you raised.
>
> We fully understand that you may be occupied with other commitments, but we would greatly value any comments you can provide on our responses before the deadline.
>
> Thank you for your attention to this matter. We eagerly look forward to hearing from you soon.
>
>
> Sincerely,
>
> 4485 Authors

---

### Official Review · Reviewer_BcQr · 2023-11-02

**Soundness:** 3 good
**Presentation:** 3 good
**Contribution:** 3 good
**Rating:** 6
**Confidence:** 4

**Summary:**

This paper presents AugLocal, a local learning method for neural networks that alleviates the backward locking problem of traditional back-prop (end-to-end) training. Another benefit of local learning is the reduced peak GPU memory utilization, as activation storage can be reused between different local layers or blocks. In AugLocal, the size of the auxiliary network attached to each local layer is controlled by how far the layer is from the output layer. This facilitates the design of auxiliary networks. The idea behind AugLocal is that gradients obtained from auxiliary networks "emulate" gradients from subsequent layers (in traditional back-prop), hence the intuition presented in this work that earlier layers require deeper auxiliary networks. The authors benchmark AugLocal against different local learning methods, as well as back-prop, on various CNNs and different datasets.

**Strengths:**

- This work tackles an important problem: local learning in neural networks.
- The paper is well-written, and ideas are clearly presented.
- The proposed idea is simple, yet effective. The idea of using gradually smaller auxiliary networks in local learning seems novel to me
- The experimental results, despite focusing on CNNs, are decently thorough, and back the authors claims.

**Weaknesses:**

- In Table 1, the authors can provide a more complete picture by including some complexity measure (similar to what is reported in Table 2 in Appendix). What is the wall-clock time of all of these methods? I suspect that AugLocal will incur some significant overhead due to having deeper auxiliary networks.
- The theoretical speedup of (d+1)/(L+1) is not very informative, as it assumes an ideal setting where all local modules can be run in parallel, which is not trivial to implement. The point of local learning is save GPU memory, which I suspect comes at the expense of extra time. It is also worth comparing AugLocal with back-prop with gradient checkpointing.


Minor comments:
- eq(2) is a bit misleading, it implies that only the auxiliary network parameters $\Phi^l$ impact the local losses, and the final loss is a function of all network parameters.

**Questions:**

- In the ImageNet experiments, the DGL numbers correspond to a much waker end-to-end baseline (66.6 vs 71.59 in this paper), thus making DGL seem weaker than it actually is. The authors already reproduced the DGL numbers (as well as other methods) for the other datasets to ensure a fair comparison. Why haven't the authors done the same for ImageNet?
- What would the GPU memory savings look like with varying batch sizes, does it require a large BS of 1024 to show significant improvement?

---

> ### Author Response · Authors · 2023-11-17
>
> Thank you for your positive recognition on our paper and helpful comments. We attach our responses and explanations corresponding to all of your comments below.
>
> **Q1:** In Table 1, the authors can provide a more complete picture by including some complexity measure (similar to what is reported in Table 2 in Appendix). What is the wall-clock time of all of these methods? I suspect that AugLocal will incur significant overhead due to having deeper auxiliary nets.
>
> **A1:** Thank you for your suggestion. As you have rightly pointed out, the wall-clock time for the proposed AugLocal depends on the specific implementation. In the case of the sequential implementation, where each hidden layer is trained sequentially one after another, AugLocal does incur increased computational overhead compared to other local learning methods. This is due to the presence of additional auxiliary networks. To provide a more complete picture, we have compared the wall-clock time across all local learning methods under the sequential implementation using ResNet-110, as shown in the table below. We have also included the results in Table 2 in Appendix. These results reveal that AugLocal (d=2) can achieve higher accuracy at the trade-off of slightly increased computational overhead in comparison to other local learning rules, and the accuracy of AugLocal can be further enhanced by leveraging deeper auxiliary nets, albeit at the expense of increased wall-clock time. Nevertheless, we would like to highlight that when having a dedicated parallel implementation, which we are still working on, the AugLocal can yield significant training speed-ups compared to end-to-end BP training, as theoretically analyzed in Appendix A.1.
>
> |  |BP|Gradient Checkpoint|PredSim|DGL|InfoPro|AugLocal(d=2)|AugLocal(d=3)|AugLocal(d=4)|AugLocal(d=5)|AugLocal(d=6)|
> |:-:|:-:|:-:|:-:|:-:|:-:|:-:|:-:|:-:|:-:|:-:|
> |GPU Memory (GB)|9.27|3.03|1.54|1.61|3.98|1.71|1.62|1.70|1.71|1.72|
> |Computational Overhead (Wall-clock Time)|-|34.1%|65.9%|76.2%|292.3%|87.2%|115.9%|135.6%|180.8%|214.6%|
> |Acc.|94.61±0.18|94.61±0.18|74.95±0.36|85.69±0.32|86.95±0.46|90.98±0.05|92.62±0.22|93.22±0.17|93.75±0.20|93.96±0.15|
>
> **Q2:** The theoretical speedup of (d+1)/(L+1) is not very informative, as it assumes an ideal setting where all local modules can be run in parallel, which is not trivial to implement. The point of local learning is save GPU memory, which I suspect comes at the expense of extra time. It is also worth comparing AugLocal with back-prop with gradient checkpointing.
>
> **A2:** We fully agree with you on this point and thank you for suggesting adding the comparison with gradient checkpointing.  As already replied in **A1**, we have updated our paper to include the wall-clock time of our method under the sequential implementation in Appendix Table 2. Moreover, to fully realize the potential of AugLocal, we are working on a GPU-based parallel implementation and we hope to provide an update in future work.
>
> Additionally, in response to the suggestion to compare AugLocal with gradient checkpointing, we have revised our paper to include this comparison in Table 4 and Appendix Table 2. Our results demonstrate that AugLocal can achieve a much lower GPU memory footprint, albeit accompanied by a moderate increase in wall-clock time.
>
> **Q3:** In the ImageNet experiments, the DGL numbers correspond to a much weaker end-to-end baseline. The authors already reproduced the DGL numbers for the other datasets. Why haven't the authors done the same for ImageNet?
>
> **A3:** Thank you for your suggestions. We have conducted the DGL experiment on ImageNet using consistent training setups. The accuracy achieved by DGL is 67.32%, which is much better than the original paper's reported accuracy of 64.4%. Our AugLocal still outperforms DGL by a large margin of 3.6% (70.92% vs. 67.32%). We have updated our paper to reflect the enhanced accuracy of DGL.
>
> **Q4:** What would the GPU memory savings look like with varying batch sizes, does it require a large BS of 1024 to show significant improvement?
>
> **A4:** Thank you for pointing this out. We have conducted experiments to analyze the GPU memory footprint of AugLocal with varying batch sizes (ranging from 128 to 2048) using ResNet-110 on CIFAR-10. As the results are given below, even with a small batch size of 128, AugLocal can achieve significant memory savings. Moreover, the memory savings further improve as the batch size increases.
>
> |Batch Size|128|256|512|1024|2048|
> |:-:|:-:|:-:|:-:|:-:|:-:|
> |GPU Memory of BP (GB)|1.18|2.34|4.65|9.27|18.51|
> |GPU Memory of AugLocal (GB)|0.38 |0.54|0.93|1.72|3.36|
> |GPU Memory Saving by AugLocal|67.8%|76.9%|80.0%|81.4%|81.8%|
>
> **Q5:** eq(2) is a bit misleading, it implies that only the auxiliary network parameters impact the local losses, and the final loss is a function of all network parameters.
>
> **A5:** Thank you for pointing this out. We have fixed it in our revised paper and please refer to it for the update.

---

### Meta-Review · Area_Chair_ScyQ · 2023-12-17

**Metareview:**

This paper explores architectures that provide local gradients to layers in a deep network via a path through an auxiliary subnetwork.  Layers in the auxiliary subnetwork share architecture with a subset of those in the primary network; the auxiliary network produces output evaluated under the same loss function as the primary network, but provides a separate backpropagation path.  After the author response and discussion, reviewer ratings range from marginal reject to accept, with three of four reviewers leaning toward accept.  The AC agrees with the majority of reviewers.  Experiments make the case that auxiliary subnetworks can assist in training to achieve similar accuracy as a baseline, while reducing GPU memory requirements.

**Justification For Why Not Higher Score:**

The overall contribution, reviewer scores, and practical impact of the technique (potentially reducing GPU memory usage during training) are consistent with acceptance as a poster.

**Justification For Why Not Lower Score:**

A clear presentation along with demonstrated experimental benefits make a case for acceptance.

---

### Decision · Program_Chairs · 2024-01-16

Accept (poster)